

# Homeotic transformations and number changes in the vertebral column of *Triturus* newts

Maja Slijepčević[1], Frietson Galis[2], Jan W. Arntzen[2] and Ana Ivanović[2,3]

[1] Institute for Biological Research "Siniša Stanković", University of Belgrade, Belgrade, Serbia
[2] Naturalis Biodiversity Center, Leiden, The Netherlands
[3] Institute of Zoology, Faculty of Biology, University of Belgrade, Belgrade, Serbia

Corresponding author
Ana Ivanović, ana@bio.bg.ac.rs

## ABSTRACT

We explored intraspecific variation in vertebral formulae, more specifically the variation in the number of thoracic vertebrae and frequencies of transitional sacral vertebrae in *Triturus* newts (Caudata: Salamandridae). Within salamandrid salamanders this monophyletic group shows the highest disparity in the number of thoracic vertebrae and considerable intraspecific variation in the number of thoracic vertebrae. *Triturus* species also differ in their ecological preferences, from predominantly terrestrial to largely aquatic. Following Geoffroy St. Hilaire's and Darwin's rule which states that structures with a large number of serially homologous repetitive elements are more variable than structures with smaller numbers, we hypothesized that the variation in vertebral formulae increases in more elongated species with a larger number of thoracic vertebrae. We furthermore hypothesized that the frequency of transitional vertebrae will be correlated with the variation in the number of thoracic vertebrae within the species. We also investigated potential effects of species hybridization on the vertebral formula. The proportion of individuals with a number of thoracic vertebrae different from the modal number and the range of variation in number of vertebrae significantly increased in species with a larger number of thoracic vertebrae. Contrary to our expectation, the frequencies of transitional vertebrae were not correlated with frequencies of change in the complete vertebrae number. The frequency of transitional sacral vertebra in hybrids did not significantly differ from that of the parental species. Such a pattern could be a result of selection pressure against transitional vertebrae and/or a bias towards the development of full vertebrae numbers. Although our data indicate relaxed selection for vertebral count changes in more elongated, aquatic species, more data on different selective pressures in species with different numbers of vertebrae in the two contrasting, terrestrial and aquatic environments are needed to test for causality.

## INTRODUCTION

The vertebral column consists of repetitive, serially homologous skeletal elements–vertebrae. Along the anterior–posterior axis, vertebrae are classified into regions,

with a conserved order and specific sizes and shapes (e.g., *Starck, 1979*). The strong re-gionalization of the vertebral column is already present early in the evolution of tetrapods (*Ahlberg, Clack & Blom, 2005*). In the early tetrapod *Ichthyostega*, five regions (cervical, thoracic, lumbar, sacral and caudal) can be recognized as in many extant amniotes.

Vertebrae develop from embryonic segments (somites) that are generated from the presomitic mesoderm in a sequential head-to tail order. This process involves a molecular oscillator, the segmentation clock, that regulates the periodicity of segment formation (*Cooke & Zeeman, 1976*; *Palmeirim et al., 1997*; *Dequéant & Pourquié, 2008*; *Gomez & Pourquié, 2009*). The duration of segment formation (somitogenesis) and the speed of the segmentation clock determine the total number of segments formed, and hence, the total number of vertebrae (*Gomez et al., 2008*; *Gomez & Pourquié, 2009*). The determination of the identity of the vertebrae (e.g., cervical or thoracic) occurs as part of the early head-to-tail patterning of the presomitic mesoderm and early somites. This head-to-tail patterning involves complex genetic mechanisms that include various signaling molecules, with an essential mediating role for the well-known Hox genes (e.g., *Dubrulle, McGrew & Pourquié, 2001*; *Diezdel-Corral et al., 2003*; *Aulehla & Pourquié, 2010*; *Mallo, Wellik & Deschamps, 2010*; *Woltering, 2012*; *Wong et al., 2015*). It is thought that the segmentation process and the head-to-tail patterning of the segments by the Hox genes can be dissociated and that this dissociation has allowed for the spectacular evolutionary diversification of vertebral formulae (*Carapuço et al., 2005*; *Gomez & Pourquié, 2009*; *Schroeter & Oates, 2010*; *Harima et al., 2013*; *Wong et al., 2015*). When there are shifts of vertebral boundaries, e.g., the cervico-thoracic boundary, these shifts involve changes in the Hox patterning of the somites along the head-to-tail axis. If indeed the segmentation process and the head-to-tail patterning of the segments are dissociated, the shifts of vertebral boundaries necessarily involve homeotic transformations of vertebrae. The involvement of homeotic transformations is further supported by the observation that in humans, xenarthra and afrotherians, intraspecific changes of the cervico-thoracic or thoraco-lumbar boundary almost always involve transitional vertebrae, i.e., partial homeotic transformations, also when the number of presacral or total vertebrae is changed (*Galis et al., 2006*; *Varela-Lasheras et al., 2011*; *Ten Broek et al., 2012*). This also confirms that initial mutations for homeotic transformations usually lead to incomplete homeotic transformations, resulting in transitional vertebral identities.

In mammals, changes in the number of cervical vertebrae are associated with deleterious pleiotropic effects that lead to selection against such homeotic transformations (*Galis & Metz, 2003*; *Varela-Lasheras et al., 2011*). In two mammalian groups (sloths and manatees) with low activity and metabolic rates, the exceptional numbers of cervical vertebrae most likely resulted from the effective breaking of pleiotropic constraints due to a relaxation of stabilizing selection against the pleiotropic effects (known as congenital abnormalities, *Varela-Lasheras et al., 2011*). Furthermore, *Galis et al. (2014)* concluded that biomechanical problems associated with initial homeotic transformations (transitional vertebrae) in fast running mammals result in strong selection against changes of the presacral vertebral counts in these species.

In other tetrapods the number of vertebrae in different regions can be more variable. Well-known examples are the variable number of cervical vertebrae in birds (*Woolfenden, 1961*) and the variable number of presacral vertebrae in squamates (*Carroll, 1997*; *Müller et al., 2010*). In both cases, these regions have a large number of vertebrae. Geoffroy St. Hilaire has postulated that as a rule—structures with a large number of serially homologous repetitive elements are more variable than structures with smaller numbers (*Geoffroy, 1832*). This notion was supported by *Darwin (1860)*. Along the same line, *Bateson (1894)* concluded that series containing large numbers of undifferentiated parts are more variable than series made up of a few, more differentiated parts.

In tailed amphibians, the presacral vertebrae vary in their number but only little in shape. A single, sacral vertebra is morphologically very similar to the vertebrae from the thoracic or trunk region, with more robust processes for attachment of the sacral ribs which are also thicker than regular thoracic ribs. There is considerable intraspecific variation in the number of thoracic vertebrae in many species of salamanders (*Adolphi, 1898*; *Gerecht, 1929*; *Peabody & Brodie, 1975*; *Jockusch, 1997*; *Litvinchuk & Borkin, 2003*). Intraspecific variation originally results from homeotic transformations that are subsequently maintained in the population. In salamanders, transitional vertebrae at the thoraco-sacral boundary have been frequently reported (*Adolphi, 1898*; *Gerecht, 1929*; *Highton, 1960*; *Worthington, 1971*; *Jockusch, 1997*; *Arntzen et al., 2015*). Such transitional vertebrae with partial thoracic and partial sacral identity result from incomplete homeotic transformations. Therefore, the frequencies of transitional vertebrae could be related to the amount of variation in the number of thoracic vertebrae within species. Data on changes in axial pattering and homeotic transformations in amphibians are relatively scarce and more data are necessary for understanding the evolution of axial pattering in amphibians and the tetrapods.

In this study we explore the relationship between variation in the number of thoracic vertebrae and transitional sacral vertebrae using eight species of the monophyletic genus *Triturus* newts as a model system. Within the family Salamandridae, which is the second most diverse group of tailed amphibians, *Triturus* newts are the most disparate in the number of thoracic vertebrae (*Arntzen et al., 2015*). *Triturus* species form a morphocline from the predominantly terrestrial *T. marmoratus* and *T. pygmaeus* with a short and stout body and 12 thoracic vertebrae to the slender and elongated, largely aquatic *T. cristatus* and *T. dobrogicus* with 15–17 thoracic vertebrae (*Arntzen, 2003*). *Triturus* species also display considerable intraspecific variation in vertebral numbers (*Gerecht, 1929*; *Crnobrnja et al., 1997*; *Arntzen et al., 2015*). Moreover, there is a well-documented, extensive hybridization in the area of sympatry of *T. marmoratus* (12 thoracic vertebrae) and *T. cristatus* (15 thoracic vertebrae). The hybridization of these two species leads to sterile F1 hybrids with intermediate morphologies and number of thoracic vertebrae (*Vallée, 1959*; *Arntzen et al., 2009*). Interspecific hybridization at contact zones also occurs between other *Triturus* species with parapatric distributions (*Mikulíček et al., 2012*; *Arntzen, Wielstra & Wallis, 2014*), providing the opportunity to investigate the relationship between vertebral number and frequencies of transitional sacral vertebrae. Here, we compared variation in the

number of thoracic vertebrae and transitional sacral vertebrae among *Triturus* species, *T. marmoratus* × *T. cristatus* F1 hybrids with parental species, and populations from contact zones with populations away from contact zones. More specifically, we explored the intra- and interspecific variation in number of the thoracic vertebrae and frequencies of transitional vertebrae at the thoraco-sacral boundary to test the following hypotheses:

(1) Species with more vertebrae in the thoracic region are more variable in the number of thoracic vertebrae than those with fewer vertebrae in the thoracic region.

(2) The higher the variation in the number of thoracic vertebrae, the higher the frequencies of transitional vertebrae are. In hybrids, we would expect that the range of variation in the number of vertebrae overlaps the ranges of parental species' variation. Also, we would expect the higher frequencies of transitional vertebrae in comparison with parental species. In species with parapatric distributions we would expect the higher variation in the number of vertebrae and the higher frequencies of transitional vertebrae in populations from contact zones in comparison with populations away from contact zones.

## MATERIALS AND METHODS

### *Triturus* newts and their characteristics

The vertebral column in *Triturus* newts is differentiated in: the cervical region—consisting of a single anterior vertebra, the atlas; the thoracic (trunk) region—consisting of a rib-bearing vertebrae; the sacral region—usually a single vertebra with well-developed stout transverse processes for the attachment of sacral ribs and pelvic girdle; the caudosacral region—up to three vertebrae that continue from the caudal to the sacral vertebra and are associated with the pelvic musculature and cloaca and the caudal region—the remaining vertebrae in the tail (Fig. 1). The body elongation in *Triturus* species appears to be correlated with the length of the aquatic phase—more terrestrial species have a short and stout trunk with relatively longer legs compared to species with a more aquatic life style which have a more elongated trunk and relatively shorter legs. Body elongation involves a larger number of thoracic vertebrae. More specifically, the number of thoracic vertebrae in the vertebral formulae varies from 12 in *T. marmoratus* and *T. pygmaeus*, which have a short aquatic phase (*T. marmoratus* only two months), 13 in *T. karelinii* and *T. ivanbureschi*, 14 in *T. macedonicus* and *T. carnifex*, 15 in *T. cristatus* to 16 or 17 in *T. dobrogicus,* the most aquatic species which has six months long aquatic phase (*Arntzen, 2003*) (Fig. 2).

The distribution of the genus *Triturus* is well documented (*Arntzen, Wielstra & Wallis, 2014*). *Triturus cristatus* and *T. marmoratus* have an area of range overlap in France and can often be found in syntopy (*Arntzen & Wallis, 1991*; *Lescure & De Massary, 2012*). Other *Triturus* species contact zones are generally narrow and show a weak but significant negative relationship between the level of hybridization and the genetic distance of species pairs (*Arntzen, Wielstra & Wallis, 2014*).

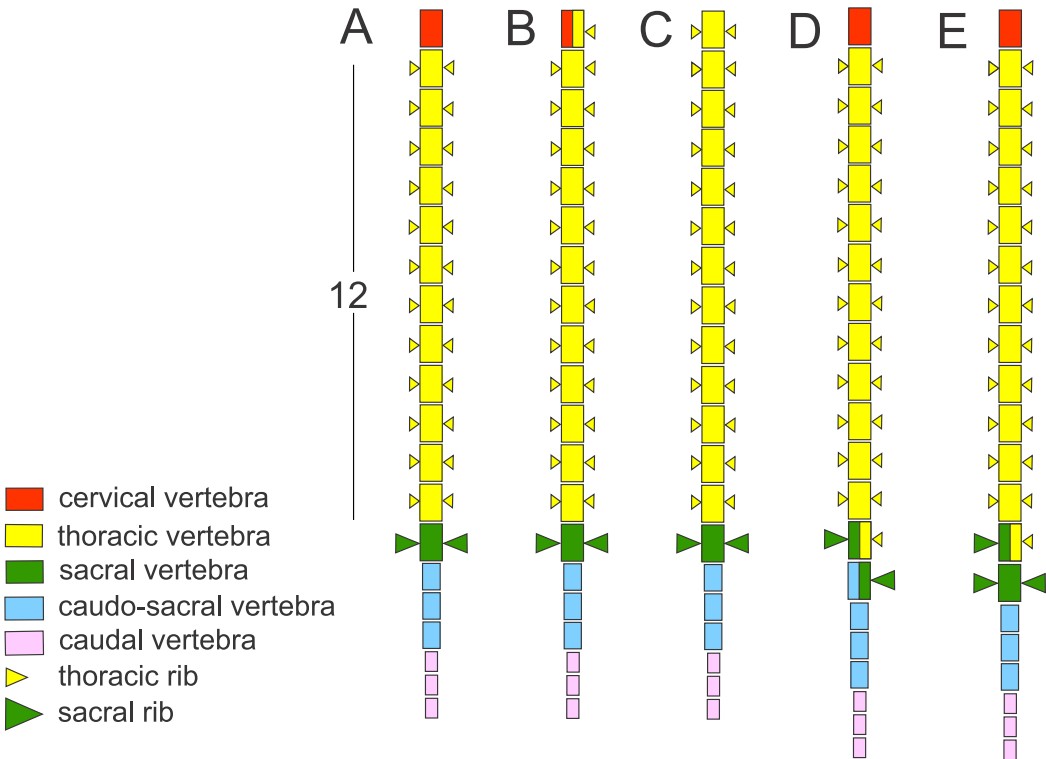

**Figure 1 Schematic presentation of the vertebral column in *Triturus* newts and homeotic transformations scored.** The regionalization of the vertebral column in *Triturus* newts and schematic presentation of scored homeotic transformations (example of *T. marmoratus*). (A) Vertebral column without homeotic transformation and regular number of vertebrae—the first three caudal vertebrae are shown; (B) incomplete homeotic transformations of cervical vertebra into thoracic; (C) complete homeotic transformation of cervical into thoracic vertebra; (D) transitional sacral vertebra with thoracic rib at one side and sacral rib at the other side followed by transitional vertebra with sacral rib at the one side and no rib at the other; (E) transitional thoraco-sacral vertebra, with thoracic rib at one side and sacral rib at the other, followed by regular sacral vertebra.

## Material analysed

We analysed axial skeletons of 1,436 adult newts that originate from 126 populations of all eight species of *Triturus* newts (Fig. 3). For this study we used X-ray images of good quality and cleared and stained skeletons. The X-ray images were obtained on a Faxitron 43855C/D with an exposure of 20–40 s at 3 mA and 70 kV. Other material was cleared with trypsin and KOH and stained with Alizarin Red S for bone deposition (*Dingerkus & Uhler, 1977*) and stored in glycerine. Analyzed specimens are from the batrachological collection of the Institute for Biological Research "Siniša Stanković", Belgrade, Serbia ($N = 601$) and from the collection of the Naturalis Biodiversity Center, Leiden, The Netherlands ($N = 835$). Our material covers the geographic, taxonomic and molecular genetic diversity of the genus. On the basis of well documented species distributions (*Mikulíček et al., 2012*; *Arntzen, Wielstra & Wallis, 2014*) populations were assigned as "central" or "fringe" based upon their geographical position away ($\geq 50$ km) or close to ($< 50$ km) congeneric species. For localities and sample size per population see Table S1.

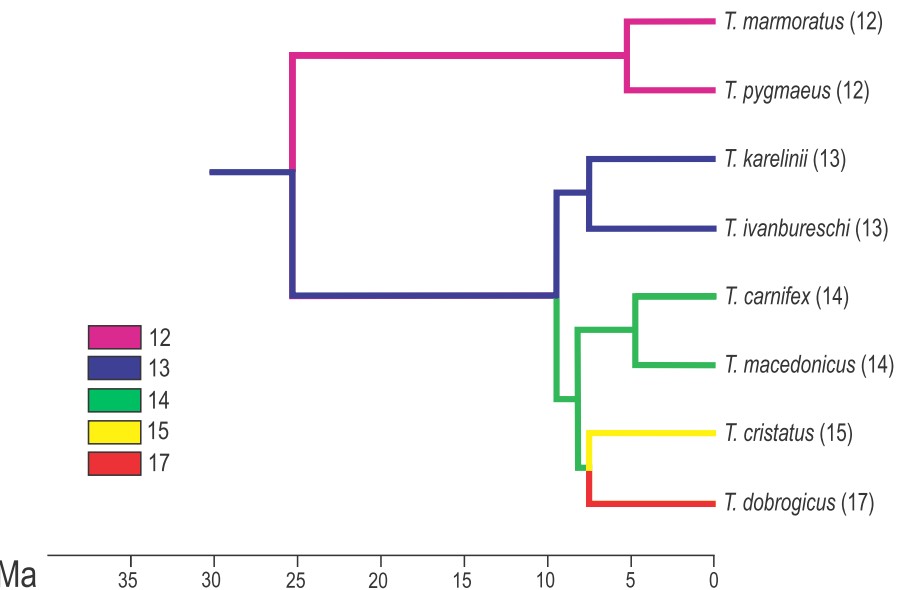

**Figure 2 Calibrated phylogeny for the genus *Triturus* with the modal number of vertebrae indicated by colour code.** Calibrated phylogeny for the genus *Triturus* with the modal number of vertebrae indicated by colour code (after *Arntzen et al., 2015*).

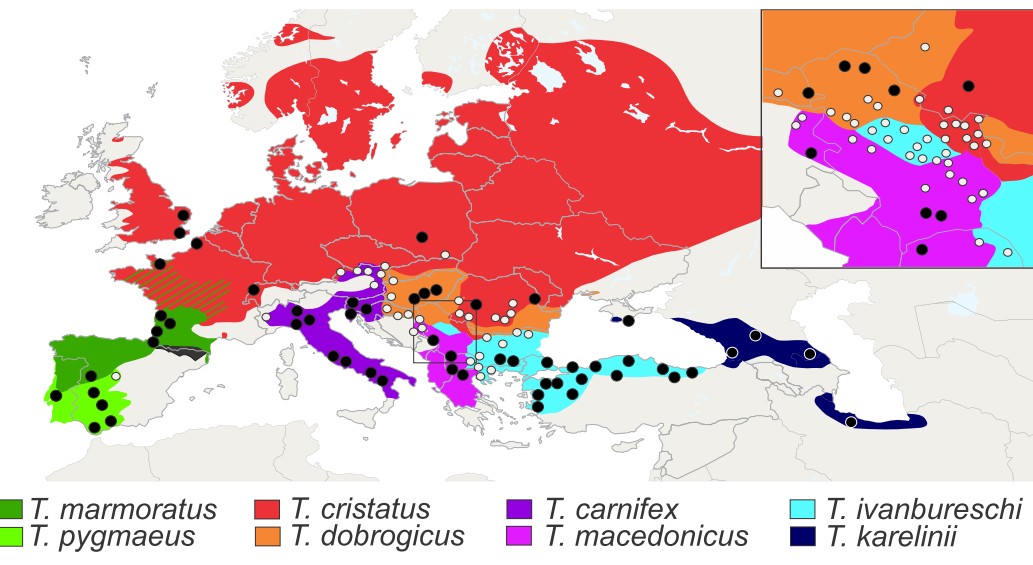

**Figure 3 Distribution of *Triturus* species and geographic positions of populations.** Distribution of eight *Triturus* species across Europe and adjacent Asia. Studied populations are marked by solid dots (central populations) and open dots (fringe populations). For detailed information see Table S1.

## Scoring vertebral formulae and transitional thoraco-sacral vertebrae

We determined the vertebral formula by counting the number of cervical (C), thoracic (T) and sacral vertebrae (S). The caudosacral and caudal regions are excluded from our formula as the detailed inspection of cleared and stained specimens showed that a variable

number of caudosacral vertebrae frequently bear small, much reduced ribs which could be fused with transverse processes and cannot always be distinguished on X-ray images. The number of tail vertebrae was available only for a subset of specimens; in most specimens tails had been removed for enzyme electrophoretic analyses or were broken or damaged.

Homeotic transformations of thoracic vertebra into sacral vertebra, or *vice versa* (transitional sacral vertebra having half of the identity of thoracic vertebra and half of the identity of sacral vertebra) were assigned 0.5 and this score was added to the number of complete thoracic vertebrae. Only complete changes of identity on one side of the vertebrae (on one side thoracic and on one side sacral) were declared transitional. Right side asymmetry of a sacral vertebra is scored when the thoracic rib is present on the right side and the sacral rib on the left side of transitional thoraco-sacral vertebra and vice versa for left side asymmetry. For a 3D model of regular and transitional thoraco-sacral vertebra obtained by CT-scanning see Data S1 and S2. We assumed that the frequency of transitional vertebrae with a complete change of identity at one side of vertebra reflects the frequency of all homeotic transformations, including more gradual ones, which could not always be scored.

## Statistical analyses

The Spearman correlation coefficient ($r_s$) was used to quantify correlation between species modal numbers of thoracic vertebrae ($T_n$) and (1) the percentage of individuals with the number of complete thoracic vertebrae different from the modal number ($T_{var}$) and (2) the range of variation in the number of thoracic vertebrae ($T_{range}$) within species. The same measure was used to quantify the relationship between percentages of transitional sacral vertebrae ($S_{tr}$) and $T_{var}$ and $T_{range}$. To test for differences between hybrids and parental species across fringe and central populations we used the *G*-test of independence. To analyse interspecific variation in a phylogenetic context, we used a well resolved time-calibrated phylogeny of genus *Triturus* (*Arntzen et al., 2015*) shown in Fig. 2. Associations derived from common ancestry were evaluated by calculating the strength of the phylogenetic signal for analysed variables ($T_n$, $T_{var}$, $T_{range}$ and $S_{tr}$). The procedure involves the random permutation of the variables over the terminal units of the phylogenetic tree (10,000 iterations), in which the test statistic is the total amount of squared change summed over all branches of the tree. We applied the phylogenetic independent contrasts approach (*Felsenstein, 1985*) to obtain a set of independent contrasts. The regression of (1) $T_{var}$ independent contrasts on $T_n$ independent contrasts and (2) $T_{range}$ independent contrasts on $T_n$ independent contrasts were used to explore the relationship between evolutionary change in the number of the thoracic vertebra in vertebral formula and amount of variation in the number of thoracic vertebrae. The regressions of $T_{var}$ and $T_{range}$ independent contrasts on $S_{tr}$ were used to explore changes and frequencies of transitional vertebrae, taking the similarity due to shared ancestry into account.

**Table 1** The number of thoracic vertebrae in *Triturus* species (central populations only).

| Species | Sample size | 12 | 12.5 | 13 | 13.5 | 14 | 14.5 | 15 | 15.5 | 16 | 16.5 | 17 | 17.5 | 18 | $S_{tr}$ (%) | $T_{var}$ (%) |
|---|---|---|---|---|---|---|---|---|---|---|---|---|---|---|---|---|
| *T. marmoratus* | 58 | **46** | 4 | 8 | | | | | | | | | | | 6.9 | 13.8 |
| *T. pygmaeus* | 55 | **52** | 1 | 2 | | | | | | | | | | | 1.8 | 3.6 |
| *T. ivanbureschi* | 175 | 1 | 3 | **150** | 4 | 17 | | | | | | | | | 4.0 | 10.3 |
| *T. karelinii* | 43 | | 1 | **40** | | 2 | | | | | | | | | 2.3 | 4.7 |
| *T. carnifex* | 66 | | | 4 | 5 | **53** | | 3 | | 1 | | | | | 7.6 | 12.1 |
| *T. macedonicus* | 67 | | | 9 | 5 | **51** | 1 | 1 | | | | | | | 9.0 | 14.9 |
| *T. cristatus* | 122 | | | 1 | | 6 | 1 | **98** | 5 | 11 | | | | | 4.9 | 14.8 |
| *T. dobrogicus* | 57 | | | | | | | 2 | | 11 | 1 | **42** | | 1 | 1.8 | 24.6 |

**Notes.**

Modal numbers of thoracic vertebrae in vertebral formulae are shown in bold. $S_{tr}$, percentage of individuals with transitional vertebrae at thoraco-sacral boundary; $T_{var}$, percentage of individuals with the complete number of thoracic vertebrae different from the modal number.

**Table 2** Overview of homeotic transformations observed in *Triturus* species. Number and percentage of individuals with transitional vertebrae are given. Left and right asymmetries of transitional sacral vertebra are shown separately.

| Species | Sample size | Transitional | % | Cervical to thoracic | | | Transitional sacral | | | Thoracic to sacral | |
|---|---|---|---|---|---|---|---|---|---|---|---|
| | | | | Complete | Incomplete | % | Left | Right | % | Incomplete | % |
| *T. marmoratus* | 58 | 5 | 8.6 | 0 | 1 | 1.7 | 1 | 3 | 6.9 | 0 | |
| *T. pygmaeus* | 55 | 1 | 1.8 | 0 | 0 | | 1 | 0 | 1.8 | 0 | |
| *T. ivanbureschi* | 361 | 25 | 6.9 | 1 | 3 | 1.1 | 6 | 12 | 5.0 | 3 | 0.8 |
| *T. karelinii* | 43 | 1 | 2.3 | 0 | 0 | | 0 | 1 | 2.3 | 0 | |
| *T. carnifex* | 123 | 8 | 6.5 | 0 | 0 | | 6 | 2 | 6.5 | 0 | |
| *T. macedonicus* | 226 | 14 | 6.2 | 0 | 1 | 0.4 | 8 | 4 | 5.3 | 1 | 0.4 |
| *T. cristatus* | 286 | 16 | 5.6 | 2 | 1 | 1.0 | 10 | 3 | 4.5 | 0 | |
| *T. dobrogicus* | 216 | 13 | 6.0 | 0 | 0 | | 7 | 6 | 6.0 | 0 | |
| Total | 1,368 | 83 | 6.1 | 3 | 6 | 0.6 | 39 | 31 | | 4 | 0.3 |

## RESULTS

### Vertebral formula and transitional sacral vertebra in *Triturus newts*

The most common vertebral formulae were 1C 12T 1S in *T. marmoratus* and *T. pygmaeus*, 1C 13T 1S in *T. karelinii* and *T. ivanbureschi*, 1C 14T 1S in *T. macedonicus* and *T. carnifex*, 1C 15T 1S in *T. cristatus* and 1C 17T 1S in *T. dobrogicus* (see Fig. 2 and Table 1). The percentage of individuals with a number of complete thoracic vertebrae different from the modal number ($T_{var}$) varied among species, from 3.6% in *T. pygmaeus* to 24.6% in *T. dobrogicus*. The range of variation in the number of thoracic vertebrae is 12–13 observed in *T. marmoratus* and *T. pygmaeus*, 12–14 in *T. ivanbureschi*, 12–14 in *T. karelinii*, 13–15 in *T. macedonicus*, 13–16 in *T. carnifex*, 13–16 in *T. cristatus* and 15–18 in *T. dobrogicus* (see Table 1). The variation in the vertebrae number per population is shown in Table S2.

Frequencies of recorded homeotic transformations in *Triturus* newt species are listed in Table 2. The least common is the homeotic transformation of cervical vertebra

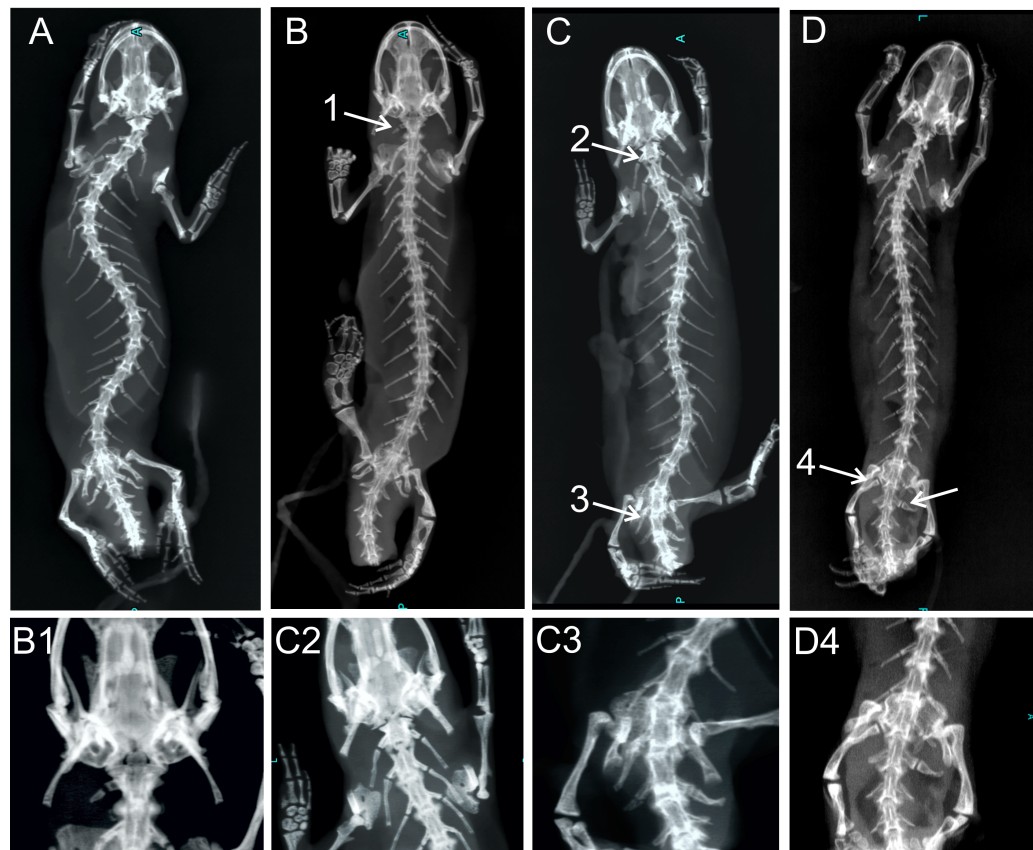

**Figure 4 X-ray images of homeotic transformations recorded.** (A) Complete vertebral column without homeotic transformations and transitional vertebrae; (B) transitional cervical vertebra (cervical into thoracic) (1); (C) complete homeotic transformation of cervical into thoracic vertebra (2) and transitional sacral vertebra with thoracic rib on the right side and sacral rib on the other side followed by sacral vertebra (3); (D) Transitional sacral vertebra—first vertebra with thoracic rib on the right side and sacral rib on the left side, followed by second transitional vertebra, with sacral rib on the right side and no rib attached on the left side (transitional sacral vertebra with a left side asymmetry) (4); Transformations are marked by arrows and numbered.

with the thoracic rib attached to one or both sides of the vertebra, recorded six times (0.41%) and in four out of eight species. Two types of transitional vertebrae at the thoraco-sacral boundary were recorded. The first type involves changes of two succeeding vertebrae—transitional sacral vertebra with thoracic rib at one side and sacral rib at the other side, followed by transitional vertebra having sacral rib at one side (opposite than previous vertebra) and no rib attached on the other side (see Figs. 1 and 4). The second type of transitional sacral vertebra involves transitional thoraco-sacral vertebra, with a thoracic rib at one side and a sacral rib at the other, followed by regular sacral vertebra. The transitional changes involving two adjacent vertebrae, thoracic and sacral (here termed transitional sacral) are more frequent than transitional changes of thoracic to sacral vertebra without changes of sacral vertebra. Excluding F1 hybrids, we recorded a transitional sacrum in 70 out of 1,368 specimens (5.1%). Both, right side and left side asymmetries were recorded (Table 2). We found that $T_n$ and $T_{var}$ are significantly

**Table 3  The number of thoracic vertebrae in *T. cristatus* × *T. marmoratus* F$_1$ hybrids and parental species.**

| Species | Sample size | Number of thoracic vertebrae | | | | | | | | | | | | | $S_{tr}$ (%) | $T_{var}$ (%) |
|---|---|---|---|---|---|---|---|---|---|---|---|---|---|---|---|---|
| | | 12 | 12.5 | 13 | 13.5 | 14 | 14.5 | 15 | 15.5 | 16 | 16.5 | 17 | 17.5 | 18 | | |
| *T. marmoratus* | 58 | **46** | 4 | 8 | | | | | | | | | | | 6.9 | 13.8 |
| *T. cristatus* | 122 | | | 1 | | 6 | 1 | **98** | 5 | 11 | | | | | 4.9 | 14.8 |
| *Hybrids* | 68 | 2 | 1 | **39** | 4 | 16 | | 6 | | | | | | | 7.4 | 35.3 |

**Notes.**

Modal numbers of thoracic vertebrae in vertebral formulae are shown in bold. $S_{tr}$, percentage of individuals with transitional vertebrae at thoraco-sacral boundary; $T_{var}$, percentage of individuals with the complete number of thoracic vertebrae different from the modal number.

positively correlated ($r_s = 0.75, p = 0.023$), indicating that species with more vertebrae in the thoracic region are more variable in the number of vertebrae. A significant correlation was also found between $T_n$ and $T_{range}$ ($r_s = 0.90, p = 0.002$), indicating that the range of variation was significantly higher in species with more thoracic vertebrae. We did not find a correlation between $S_{tr}$ and $T_{var}$ ($r_s = 0.31, p = 0.46$) or between $S_{tr}$ and $T_{range}$ ($r_s = 0.13, p = 0.76$).

## Phylogenetic comparative analyses

We found a statistically significant phylogenetic signal in $T_n$ ($p = 0.013$) and $T_{range}$ ($p = 0.033$) and no significant phylogenetic signal in $T_{var}$ ($p = 0.730$) and $S_{tr}$ ($p = 0.970$). The regression of (1) $T_{var}$ independent contrasts on $T_n$ independent contrasts ($p = 0.018$) and (2) the regression of $T_{range}$ independent contrasts on $T_n$ independent contrasts revealed a significant relationship between the increase in the number of thoracic vertebrae and the amount of variation in the number of vertebrae ($p = 0.006$). We found no significant relationship between $T_{range}$ independent contrasts and $S_{tr}$ independent contrasts ($p = 0.413$).

## Hybridization and variation in vertebral formula

There were statistically significant differences in changes in vertebral formulae between central' and fringe populations (*G*-test for independence, $G = 18.61, p = 0.001$). For fringe populations, the observed range of variation in number of thoracic vertebrae is 12–15 in *T. ivanbureschi*, 13–16 in *T. macedonicus*, 13–15 in *T. carnifex*, 13–17 in *T. cristatus* and 14–18 in *T. dobrogicus*. In *T. dobrogicus* and *T. ivanbureschi* fringe populations differed significantly from central populations in the frequencies of individuals with non-modal vertebrae formulae. For other species no significant differences between central and fringe populations were found (Table 4). In *T. cristatus* × *T. marmoratus*, sixty F1 hybrids (88.2%) have a vertebral formula with an intermediate number of thoracic vertebrae (Table 3). Six hybrids (8.8%) possess an incomplete homeotic transformation. Among these, one has an incomplete transformation of a cervical into a thoracic vertebra. The most frequent incomplete homeotic transformation involves an asymmetrical sacrum. The frequencies of transitional sacral vertebra in hybrids and parental species are similar (*G*-test for independence, $G = 1.07, p = 0.59$).

**Table 4 The number of individuals with regular and changed vertebral formulae in "central" and "fringe" populations of species with parapatric distributions.**

| Species | Number with modal vertebral formula | | Number with non-modal vertebral formula | | G | p |
|---|---|---|---|---|---|---|
| | Central | Fringe | Central | Fringe | | |
| T. ivanbureschi | 150 | 110 | 25 | 76 | 18.86 | *** |
| T. carnifex | 53 | 37 | 13 | 20 | 2.09 | ns |
| T. macedonicus | 51 | 122 | 16 | 37 | 0.006 | ns |
| T. cristatus | 98 | 130 | 24 | 34 | 0.032 | ns |
| T. dobrogicus | 42 | 67 | 15 | 92 | 6.65 | ** |
| Total | 394 | 466 | 83 | 259 | | |

**Notes.**

ns, not significant.

** $p < 0.01$.

*** $p < 0.001$.

## DISCUSSION

Our study shows a substantial variation in the number of thoracic vertebrae in *Triturus* newts, suggesting the absence of strong selection against change in the number of thoracic vertebrae. In agreement with the postulations of Geoffroy St. Hilaire, Darwin and Bateson, the variation in the vertebral column is positively correlated with the number of vertebrae. Using independent contrasts we observed a statistically significant, correlated evolutionary change between an increase in the modal number of thoracic vertebrae and variation in vertebrae number. The range of variation in the number of thoracic vertebrae is also significantly higher in species with a larger modal number of thoracic vertebrae. In *T. marmoratus* and *T. pygmaeus* the variation is limited to one extra thoracic vertebra, while in crested newt species the variation range is up to four vertebrae. Although such a pattern of intraspecific variation is in agreement with Geoffroy St. Hilaire's and Darwin's rules, the observed pattern may also represent an association between variation in vertebral number and differences in selection in different habitats that *Triturus* species occupy, in particular the amount of time they spend in the terrestrial versus the aquatic environment. During the terrestrial phase the limbs support the weight of the body and provide forward propulsion by the synchronous use of diagonal limb pairs. In water where the body weight does not need to be supported by the limbs the newts move by tail propulsion with the limbs tightly held against the body to reduce hydrodynamic drag (*Gvoždik & Van Damme, 2006*). Therefore, the selection pressures related to specific biomechanical requirements are probably different with respect to the duration of the terrestrial and aquatic phase. The larger variation in the number of thoracic vertebrae may indicate relaxed selection in more aquatic species, but more data on different selective pressures in terrestrial versus aquatic environments are needed to find out whether this is the case.

### Frequencies of transitional vertebrae

In *Triturus* newts, the frequency of transitional changes at the cervico-thoracic boundary is more than ten times lower than changes at the thoraco-sacral boundary. This is also

observed in other salamanders (*Wake & Lawson, 1973*) and mammals (*Galis et al., 2006*). This pattern may be explained by stronger interactivity and low modularity of developmental processes during the early organogenesis, or phylotypic stage, when the cervical vertebra is determined (*Galis et al., 2006*). At later stages, development is increasingly less interactive and more modular, such that changes are expected to be associated with fewer pleiotropic effects. The hypothesis that mutations with an effect during early organogenesis stage lead to more pleiotropic effects and as a consequence to more vulnerability and mortality than earlier or later stages was tested and strongly supported in rodents (*Galis & Metz, 2001*). In amphibians indirect support for this hypothesis is discussed by *Galis, Wagner & Jockusch (2003)*. We do not know the cause of the constraint on the number of cervical and sacral vertebra in tailed amphibians, but further studies in various amphibian groups that will consider survival rates of individuals with changes in the cervical and sacral region across ontogenetic stages should provide valuable data to solve this issue.

Although we hypothesized that frequencies of transitional vertebrae at the thoraco-sacral boundary should be correlated to the range of variation in the number of thoracic vertebrae as in mammals (*Ten Broek et al., 2012*), no correlation was found. Available literature data indicate that incomplete homeotic transformation of sacral vertebrae are relatively common, with up to 10% across the various salamander lineages: 4.5% in *Batrachoseps attenuatus* (*Jockusch, 1997*), 5.7% in *Rhyacotriton olympicus* (*Worthington, 1971*), 6% in *Plethodon cinereus* (*Highton, 1960*), up to 9% for newt genera *Lissotriton* and *Ichthyosaura* (*Arntzen et al., 2015*) and between 1.9% and 9.0% in *Triturus* newts (this study). The lower than expected incidence of transitional vertebrae could result from developmental mechanisms favoring complete numbers of thoracic vertebrae and/or from selection against transitional sacral vertebrae due to associated problems related to an asymmetric sacrum (c.f. *Galis et al., 2014*). Potential problems associated to asymmetrical sacrum might arise due to asymmetrical muscle attachments, blood vessels and innervation, or biomechanical problems during locomotion. In salamanders, the selection pressures related to specific biomechanical requirements are probably different in fully aquatic larvae and metamorphosed individuals that spend most of their time on land. Furthermore, selection pressures may vary with respect to the duration of annual aquatic and terrestrial phase. More detailed morphological and functional studies of locomotion of larval and metamorphic stages could shed more light on the functional significance of variation in the axial skeleton in *Triturus* newts. However, it is possible that our results are biased as we have not included the full range of transitional vertebrae. We scored only easily identifiable transitional vertebrae with complete morphological transformations of one side of the vertebra under the assumption that the frequency of these transitional vertebrae reflects the total amount of homeotic transformations. Nonetheless, initial mutations for homeotic transformations can lead to a whole series of gradually transitional homeotic transformations; in the case of thoraco-sacral vertebrae ranging from predominantly thoracic and only slightly sacral to predominantly sacral and slightly thoracic. Inclusion of all transitional vertebral morphologies might change the observed relationship between

incomplete homeotic transformations and changes in the number of thoracic vertebrae in newts.

## Hybridization, marginality and homeotic transformations

Hybridization and marginality significantly increase variability in the number of thoracic vertebrae but there is no change in the frequency of transitional vertebrae. Crosses between *T. cristatus* (15 vertebrae, range 13–16) and *T. marmoratus* (12 vertebrae, range 12–13) produced phenotypes with 13 thoracic vertebrae, an intermediate number. It is interesting to note that 13 thoracic vertebrae is the only number that is shared by both parental species. In *T. cristatus* × *T. marmoratus* offspring there is considerable mortality and almost all of F1 hybrids (∼90%) had *T. cristatus* as mother. The *marmoratus*-mothered specimens were all male, due to low survival of female embryos (*Arntzen et al., 2009*). Developmental anomalies in *T. cristatus* × *T. marmoratus* crosses, including more digital anomalies compared with parental species (hybrids 16.9%, parental species pooled 5.4%) (*Vallée, 1959*; more data in *Arntzen & Wallis, 1991*) are observed, and therefore, the higher number of changes in the axial skeleton may be related to a generally higher number of anomalies. The high mortality may also influence the incidence of the variability and transitional vertebrae.

Significantly higher frequency of changes in vertebral formula in fringe populations of *T. ivanbureschi* and *T. dobrogicus* species may well have to do with the confirmed presence of hybridization in the contact zones of *T. cristatus* and *T. dobrogicus* populations (*Mikulíček et al., 2012*), of *T. carnifex* and *T. dobrogicus* populations (*Wallis & Arntzen, 1989*) and of *T. ivanbureschi* and *T. macedonicus* populations (*Arntzen, Wielstra & Wallis, 2014*). However, the effect of the various genotype combinations on the survival rate and morphology of these species remains to be studied.

In conclusion, *Triturus* newts have a relatively large amount of variation in the number of thoracic vertebrae, both with respect to the frequency of non-modal numbers and the range of variation. In agreement with Geoffroy St. Hilaire's rule, variation was larger in species with a larger number of thoracic vertebrae. The absence of a correlation between the frequency of homeotic change (transitional sacral vertebrae, $S_{tr}$) and variation in the number of vertebrae ($T_{var}$, $T_{range}$) could be a result of developmental mechanisms that favour complete numbers of presacral vertebrae and/or selection against transitional vertebrae in this group of tailed amphibians.

## ACKNOWLEDGEMENTS

We thank Hans Metz and Joost Woltering for discussions, Ben Wielstra for providing a distribution map and Marieke Vinkenoog for help with X-ray imaging.

### Funding

This work was supported by the Serbian Ministry of Education and Science (grant no. 173043), grants from SyntheSys (NL-TAF 1245, NL-TAF 3082) and a Naturalis Biodiversity

Center 'Temminck fellowship'. The funders had no role in study design, data collection and analysis, decision to publish, or preparation of the manuscript.

## Grant Disclosures

The following grant information was disclosed by the authors:

Serbian Ministry of Education and Science: 173043.

SyntheSys: NL-TAF 1245, NL-TAF 3082.

Naturalis Biodiversity Center 'Temminck fellowship'.

## Competing Interests

The authors declare there are no competing interests.

## Author Contributions

- Maja Slijepčević performed the experiments, analyzed the data, contributed reagents/materials/analysis tools, wrote the paper, prepared figures and/or tables, reviewed drafts of the paper.
- Frietson Galis conceived and designed the experiments, wrote the paper, reviewed drafts of the paper.
- Jan W. Arntzen conceived and designed the experiments, contributed reagents/materials/analysis tools, wrote the paper, prepared figures and/or tables, reviewed drafts of the paper.
- Ana Ivanović conceived and designed the experiments, analyzed the data, contributed reagents/materials/analysis tools, wrote the paper, prepared figures and/or tables, reviewed drafts of the paper.

## Data Availabilitiy

The raw data are provided in Table S2.

## Supplemental Information

Supplemental information for this article can be found online at http://dx.doi.org/10.7717/peerj.1397#supplemental-information.

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
