# Peer review of "Homeotic transformations and number changes in the vertebral column of Triturus newts"

_PeerJ, doi:10.7717/peerj.1397_

## Round 0.1 · original submission · Minor Revisions

· Academic Editor

Minor Revisions

The reviewers both liked the paper and see only minor revisions needed- if these are detailed and include a point-by-point Response to Reviewers document, I may be able to accept this MS without further review. Congratulations on a warm reception to this paper! I agree it is interesting and valuable.

Reviewer 1 ·

Basic reporting

- There are frequent mistakes in the English, particularly the absence of articles. I think this may result from the (presumed) native language of the first and last author, which uses cases for this purpose as I came to understand. I did not find time to compile a complete list and also it would not always be clear if the authors mean to use "a" or "the". I think that one of the Dutch authors on the paper would easily identify these mistakes.
- lines 227-229 inconsistent use of ( )
- line 306 "shed" rather than "throw"

Experimental design

No comments

Validity of the findings

I think that the assumption that "changes in the number and
identity of the vertebrae necessarily involve homeotic transformations, except in the last-formed tail region" (line 52, 53) is flawed. Changes in number of vertebrae can arise through changes in the speed of the somitogenesis clock, meaning that a certain part of the body axis simply generates more vertebrae (Gomez et al. 2008 doi:10.1038/nature07020, Gomez and Pourquie 2009 DOI: 10.1002/jez.b.21305).
The authors therefore should distinguish better between the possibility for true homeotic transformation by for instance shifts in the Hox expression domains and changes in the number of somites formed within a certain homeotic ´Hox zone´. I think it is important if this aspect is mentioned because it can have significant implications for how the authors interpret their dataset:

- Regarding the scarcity of cervical to thoracic transformations this point may play a role too. If we would for instance assume an animal with a 10% deviation of the somitogenesis clock, then on every 10 vertebrae an additional vertebra would form. Clearly this would lead to at least one more thoracic vertebra but not to a cervical vertebra. In this sense one can imagine that Geoffroy St. Hilaire´s rule of variation also applies to the different regions of the axial formula, i.e., the number of thoracic vertebrae is more variable because there are more. Slight variations in the speed of the somitogenesis clock will accumulate to form an additional vertebrae within a certain homeotic domain, while this effect will be negligible for the cervical region. The same is true for the sacral region (as I understand the authors find frequent partial transformations, but not multiple sacral vertebrae). The presence of transitional vertebrae at the sacral but not cervical end would be the result of the fact that the somites are formed in artero-posterior direction. Meaning that slight differences in speed of the somitogenesis clock would only become visible in the caudal end of the axis.

- Transitional vertebrae might result from an offset between homeotic zones (Hox gene domains) and the somitogenesis clock, such that for instance a somite forms exactly on the anterior boundary of the Hox11 expression domain. This would explain the lack of correlation between frequency of transitional vertebrae and change of the total number of vertebrae. In fact one may even expect an inverse correlation, as a faster somitogenesis clock would lead to smaller somites and therefore a smaller chance of somites coinciding with a critical homeotic boundary resulting in a transitional vertebrae.

As conclusion: I would recommend the authors to consider the somitogenesis clock as source of changes in vertebrae number and identity (together with the possibility for homeotic transformations coming from changes in Hox zones). This is no criticism on the data set which I think is robuust. It just would make their paper more interesting.

Additional comments

A nice paper, I will be happy to see it published. I think that small changes incorporating some of the suggestions into the interpretation and discussion of the dataset would make the paper more interesting (hence more read and more cited). I advise minor revisions (i.e. without re-review). However, may you, after implementing some of the above comments, want feedback on them, I suppose you can ask the editor to send it out again.

Reviewer 2 ·

Basic reporting

No Comments

Experimental design

No Comments

Validity of the findings

No comments

Additional comments

While doing my masters on skeletal development in anurans I also came across several ‘homeotic transformations’ within the axial skeleton. This intrigued me with a few samples and am excited about the large dataset provided by these authors as well as the dissection of variation from individual species datasets to look at central vs peripheral range variation. The icing on the cake is also looking at effects of hybridization. There are several aspects of the morphological analysis that merit further study at both a comparative morphological level as well as gene patterning. I have several grammatical, minor, comments (listed in the attached .doc file) and have few major comments.

Annotated reviews are not available for download in order to protect the identity of reviewers who chose to remain anonymous.

---

## Round 0.2 · accepted · Accept

· Academic Editor

Accept

The revisions are clearly satisfactory; essentially all reviewer critiques have been adopted into the revised MS, producing a superior paper. I can now easily accept the study- congratulations!

Please ensure that the maximal amount of supplementary/archived raw data are made available to satisfy PeerJ's criteria for openness and reproducibility of scientific research.